# GMesh: A Flexible Voronoi-Based Mesh Generator with Local Refinement for Watershed Hydrological Modeling

Nicolás Velásquez [1],*, Miguel Díaz [2] and Antonio Arenas [2]

1 Florida Institute of Technology, Mechanical and Civil Engineering Department, Melbourne, FL 32901, USA
2 Department of Civil, Construction and Environmental Engineering, Iowa State University, Town Engineering Building 813 Bissell Road, Ames, IA 50011, USA; adi10136@iastate.edu (M.D.); aarenas@iastate.edu (A.A.)
* Correspondence: nvelasquez@fit.edu; Tel.: +1-321-6747119

**Abstract**

Partial Differential Equation (PDE)-based hydrologic models demand extensive preprocessing, creating a bottleneck and slowing down the model setup process. Mesh generation typically lacks integration with hydrological features like river networks. We present GHOST Mesh (GMesh), an automated, watershed-oriented mesh generator built within the Watershed Modeling Framework (WMF), to address this. While primarily designed for the GHOST hydrological model, GMesh's functionalities can be adapted for other models. GMesh enables rapid mesh generation in Python by incorporating Digital Elevation Models (DEMs), flow direction maps, network topology, and online services. The software creates Voronoi polygons that maintain connectivity between river segments and surrounding hillslopes, ensuring accurate surface–subsurface interaction representation. Key features include customizable mesh generation and variable refinement to target specific watershed areas. We applied GMesh to Iowa's Bear Creek watershed, generating meshes from 10,000 to 30,000 elements and analyzing their effects on simulated stream flows. Results show that higher mesh resolutions enhance peak flow predictions and reduce response time discrepancies, while local refinements improve model performance with minimal additional computation. GMesh's open-source nature streamlines mesh generation, offering researchers an efficient solution for hydrological analysis and model configuration testing.

**Keywords:** mesh generation; PDE-based hydrological models; flooding

## 1. Introduction

Freeze et al. introduced the blueprint for physically based digitally simulated watershed models [1]. Later, Politano et al. described their main features, highlighting their focus on surface–subsurface interactions relying on differential equations solved over a computational mesh [2]. In addition to the significant data for parameterization, the mesh generation process is time-consuming and tedious when not fully automated. These factors partly explain why physically based watershed modeling has not become widely used in hydrology despite some improvements in recent years [3]. Manual mesh generation can often take significant time, ranging from days to weeks, depending on the complexity of the watershed and the spatial resolution required, creating substantial difficulties in modeling workflows. Similar challenges have been documented in hydrological modeling and other computational domains [4,5]. Challenges with mesh generation processes are not unique to hydrology; other disciplines that rely on mesh-based simulations have identified similar problems. For example, a report by NASA on computational aero science

states that "mesh generation and adaptivity continue to be significant bottlenecks in the Computational Fluid Dynamics workflow [6]." In hydrological modeling, the accuracy and computing time heavily depend on the mesh and the network used to represent the watershed [7,8]. However, there is still a gap around mesh generation that considers the connectivity between the network and hillslope elements [9].

Several mesh generation tools have been developed for hydrologic modeling. Some examples include the Triangle Software [10], HydroGeoSphere [11], Hydrus3D [12], and Gmsh [13]. Nevertheless, most of these tools require intensive manual intervention or editing of several configuration files. In the case of Hydrus3D and Delft3D [14], most of the work uses graphical user interfaces (GUIs), which facilitate the work but, at the same time, impose configuration and automation limitations along with paywalls. Moreover, most tools have limited options (including paywalls) when it comes to generating mesh refinements at specific localizations within the watersheds.

GMesh was developed to fill the described gaps, allowing modelers to quickly set up a mesh for a watershed using the Digital Elevation Model as a starting point and setting up configurations with variable resolution. The need for an automated, flexible mesh generator capable of handling watershed-scale simulations has become increasingly evident as the complexity and scope of hydrological projects continue to grow [15,16]. GMesh was designed to address those challenges in hydrological modeling. We built GMesh on top of the Watershed Modeling Framework (WMF) [17], a Python interface that allows hydrologic analysis and simulation using code. This feature enables users to automate the mesh generation process while exploring multiple configurations. Along with the WMF, GMesh allows the generation of meshes with different configurations, the specification of refinement areas, and changes in the network threshold definition, and automatically assigns properties to the generated polygons. This paper presents the development of GMesh and its implementation within the GHOST hydrological model.

The structure of this paper is as follows: In Section 2, we describe the architecture and the steps GMesh took for the mesh generation. Section 3 presents examples and implementations of GMesh along with GHOST. Finally, in Section 4, we present our conclusions and future work.

## 2. Materials and Methods

The GHOST mesh generator (GMesh) has been developed as a class of the Watershed Modeling Framework (WMF) [17]. The WMF is a Fortran–Python module designed to provide tools to perform hydrological analysis and modeling that conceptualizes the watershed as an object with a defined topology, properties, and functions. As shown in Velasquez et al. [17], with the WMF, a Python user can quickly delineate a watershed and extract several characteristics of it, including the required files to run the TETIS distributed hydrological model [18,19] and the Hillslope Link Model (HLM) [20]. GMesh constitutes an additional set of tools that allows us to set up the mesh and files required by GHOST [2]. The following section describes the GMesh architecture and its main functions and usage.

### 2.1. GMesh Architecture

GMesh was developed on top of the watershed cell structure defined by the WMF. GMesh execution requires a Digital Elevation Model (DEM) and a single-direction map (D8) [21]. Also, the watershed structure must have a defined accumulated area threshold for the starting point of channels. Once the three described elements are determined, GMesh generates the channel network segments' topology and the corresponding mesh points, computes the Voronoi polygons, and defines the topological connection between them. Figure 1 presents a summary of the steps described. Following, we explain each step in

detail, and in the summary section, we present the links to the GitHub repositories and the data used for this work.

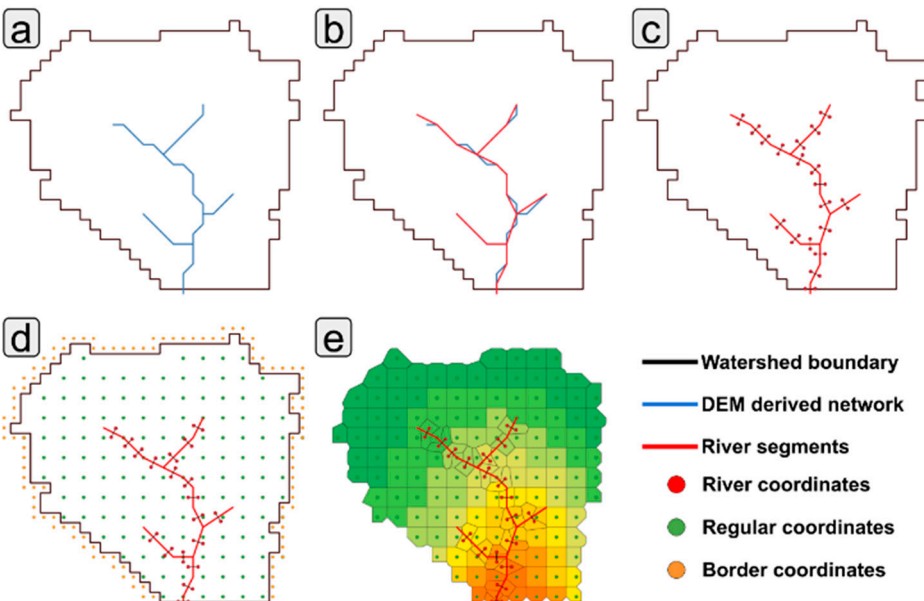

**Figure 1.** Steps followed by GMesh to generate the river segments (red network) and the watershed Voronoi polygons (colored mesh on panel (**e**)). The procedure starts by delineating the watershed (black envelope) and its network (blue lines in (**a**,**b**)) using WMF (**a**). Then, it generates the segment network (red in (**b**)) and the coordinates surrounding it (red dots and lines in (**c**)). Finally, GMesh adds the regular grid (green dots in (**d**)) and border (orange dots in (**d**)) coordinates (**d**) and generates the polygons (colored polygons in (**e**)).

### 2.1.1. Network Pre-Processing

The network is the primary driver of GMesh. As a first step, GMesh obtains a version of the network in which each channel's reach is reduced into segments with the same or less complexity in geometry. Then, GMesh defines the segment's connectivity from upstream to downstream using the network connectivity defined by the WMF. GMesh also checks for elevation coherence between upstream and downstream segments in this process. In this process, GMesh adjusts river segment elevations, ensuring that the downstream elevation of each segment is lower than the upstream one. We show the described process in Figure 1b.

### 2.1.2. Mesh Generation

GMesh defines the coordinates of the points that will generate the mesh based on the network segments and the user-defined grid size. The first coordinates of the mesh correspond to points surrounding the network segments. At every segment, GMesh defines coordinates over lines perpendicular to the segment. The distance between perpendicular lines and the length of each line defines the density of points surrounding the network. The user can adjust both parameters. Then, GMesh removes overlapping or too-close points using a threshold minimum distance. At the end of this process, GMesh obtains a collection of points around the segments (Figure 1c).

Once the segment points are defined, GMesh proceeds to populate the points collection with a regular grid of pre-defined distance and removes the ones close to the segment points (green points in Figure 1d). Finally, GMesh uses a dilation process on the WMF watershed definition to obtain the borderline points (orange points in Figure 1d).

### 2.1.3. Polygon Definition

With the mesh points defined, GMesh computes the Voronoi polygons and extracts their connectivity. From the polygons, GMesh differentiates between valid elements (river network and regular grid) and invalid ones (created from border points). GMesh eliminates the polygons that belong to the invalid mesh points or have no valid neighbors (Figure 1e). This process also determines the neighbors of each polygon, which is essential for lateral processes in hydrological models.

After the network and mesh generation process is complete, GMesh determines each river segment's left and right polygons. These polygons indicate the hillslope portions that drain directly into the channels. Finally, GMesh checks for elevation correctness and reduces the number of faces of the polygons that are over the watershed boundary.

At the end of the described steps, GMesh can write the input files required by the hydrological model. The input files include a mesh file indicating the polygon faces, their connectivity, and properties, a river file with the segment's topology and properties, and the vectorial files corresponding to both. Also, GMesh offers the option to retrieve the polygon's soil and land use properties using the Google Earth Engine (GEE) API for Python (for this option, the user must set up an account). Polygons and segment definitions are stored in geopanda dataframes, making the adaptation of GMesh straightforward to other models.

### 2.1.4. Localized Mesh Refinement

In addition to the previously described process, GMesh allows mesh refinement in areas of interest, increasing the simulation detail at specific locations. The user should provide an additional raster layer and a dictionary indicating the categories and their properties to use this option. The properties include: the distance between the points of the grid mesh, the distance between perpendicular lines along the river segments, and the length of these lines. Refinement is highly useful for inquiring about specific processes and improving models' performance [22,23]. The results section presents an example of a mesh generated using the local refinement option.

### 2.2. Functions and Usage

The previously described steps can be found in the *ghost_preprocess* class. As mentioned before, the class uses the WMF watershed class as a primary input, the path to the DEM, the specified length of the network segments, and, optionally, the refinement parameters. After its definition, the user can iterate over the following class functions:

- *Get_segments_topology*: Obtains the connectivity between the new channel segments (Figure 1b).
- *Get_mesh_river_points*: Obtains the coordinates of the points surrounding each network segment. It extracts two points per segment, one over the left and the other over the right (Figure 1c).
- *Get_mesh_grid_points*: Defines the coordinates of a regular grid used to populate the mesh inside the watershed (Figure 1d).
- *Get_vornoi_polygons*: Derives the polygons using the river, regular grid, and border mesh points. Also, it differentiates between them.
- *Define_polygons_topology*: Defines the valid polygons, the connectivity between them, their connectivity with the segments network, and their properties (Figure 1e).

The described functions can be used in series or independently, allowing easy changing of the segments and polygon configuration for a given watershed. The refinement option can be activated when setting up the project in *ghost_preprocess* using a dictionary that indicates the key that identifies a refinement area and its parameters.

### 2.3. Data and Regions of Implementation

Using the WMF preprocessor along with GMesh, we set up finite element discretization for three watersheds (see Figure 2): Bear Creek and Iowa Creek in Iowa, USA (with areas of around 82 km$^2$ and 533 km$^2$, respectively), and La Maria Creek in Medellín, Colombia (with an area of around 62 km$^2$). The generated meshes were tested by performing GHOST simulations using different options and configurations, changing the detail level, and introducing local refinements. GHOST was tested with the mesh generated under various configurations and using North American Land Data Assimilation System (NLDAS) rainfall and temperature [24] as the main forcings. Additionally, we used the information from the Iowa Creek U.S. Geological Survey (USGS) discharge gauge 05470500. We accessed both datasets on 20 May 2024. The following results describe the steps and results obtained.

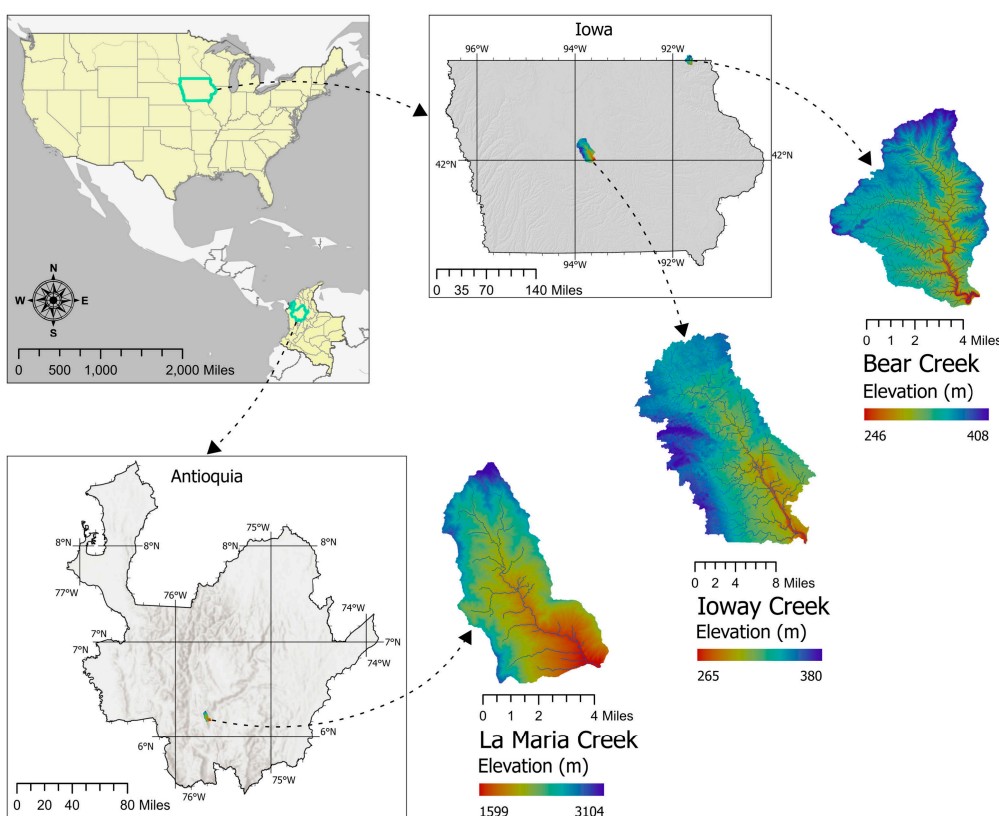

**Figure 2.** From top to bottom: Localization and description of Bear Creek, Iowa Creek (in Iowa, USA), and La Maria Creek (in Medellín, Colombia).

### 2.3.1. Bear Creek

This example presents a mesh generated for the Bear Creek watershed (top in Figure 2) using a DEM of 30 m and its corresponding D8 map (Figure 3a and b, respectively). The DEM was resampled from the USGS 1/3 national Digital Elevation Model [25] and was accessed on 15 March 2024. Bear Creek is a tributary of the Upper Iowa River. The watershed encompasses a landscape characterized by steep terrain prone to erosion. This susceptibility has led to concerns over excessive soil erosion affecting the basin's croplands, pastures, and forests. The Bear Creek Watershed Project has implemented efforts to mitigate these issues, aiming to enhance water quality by reducing pollutants such as ammoniated manure. Additionally, the creek frequently experiences flooding, exacerbating erosion and sediment transport, posing challenges for water management in the region. GHOST was run using NLDAS data as the primary forcing.

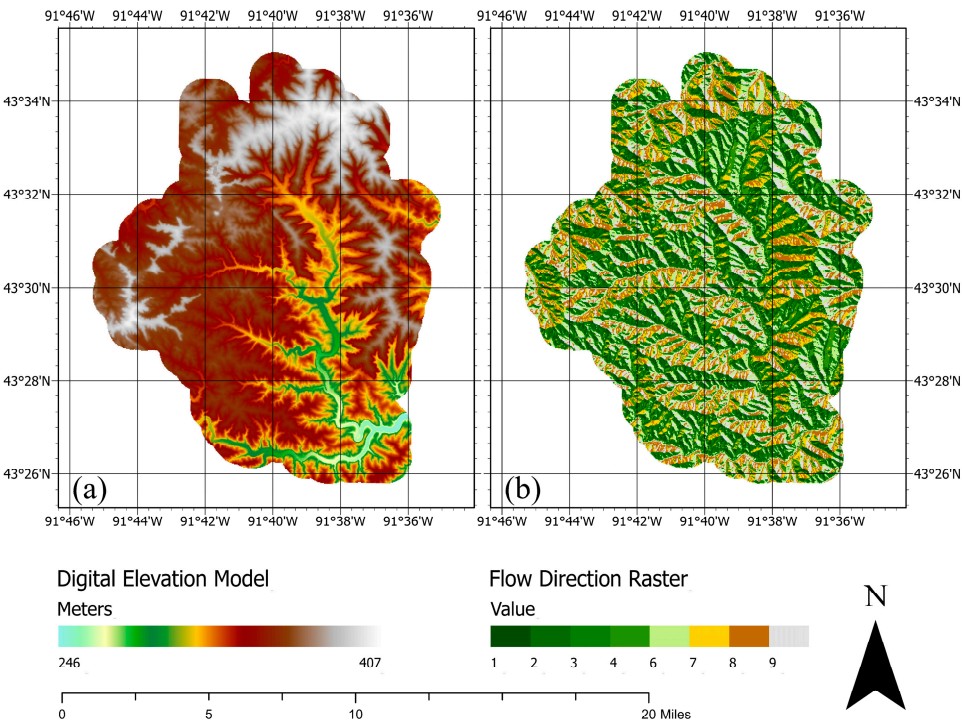

**Figure 3.** Digital elevation model (**a**) and flow direction map (**b**) for Bear Creek, Iowa.

### 2.3.2. Iowa Creek

With 553 km$^2$ and a different landscape, the Iowa Creek covers a larger domain. The DEM for this watershed corresponds to the same dataset used for Bear Creek. The watershed develops over the Des Moines Lobe till plain. Land use is predominantly corn–soy row-crop agriculture with urban pockets. Soils are mainly Clarion–Nicollet–Webster loam to clay loam, poorly drained in depressions and extensively tile-drained. The landscape mixes prairie-derived uplands, closed depressions/potholes, and sandy–silty riparian floodplains. Hydrologically, the system is highly altered and flashy: rapid runoff produces large peak-to-baseflow ratios, intermittent tributaries, and nutrient pulses during storms.

### 2.3.3. La Maria Creek

Located in Medellín, Colombia, La Maria Creek is a smaller watershed (62 km$^2$) developed in a steep topographic terrain. The DEM of La Maria has a resolution of 5 m and was obtained from the Area Metropolitana del Valle de Aburra (AMVA). The topography exhibits an elevation difference of around 1500 m between the watershed outlet and its divisor. Moreover, urban development and protected areas dominate the watershed land use. La Maria was included to present the capability of GMesh in rough terrains using a DEM with a resolution of 5 m.

## 3. Results

### 3.1. GMesh over Three Different Watersheds

The following section summarizes GMesh implementation over the Bear Creek, Iowa Creek, and La Maria Creek watersheds. For each case, we followed the steps described in Section 2.1. Table 1 presents the summary of the mesh generation, including the number of elements, the generation time, the average number of faces, and the maximum number of faces. According to the results, generation time is highly related to the watershed area. The Iowa Creek watershed took around one hour, while La Maria took less than 2 min.

Nevertheless, Gmesh exhibited stability, deriving meshes with similar properties in the three cases in their polygons.

**Table 1.** Computational time required to generate the mesh as a function of the basin area.

| Watershed | Area (square km) | Number of Elements | Generation Time (min) | Average Number of Faces | Max Number of Faces |
|---|---|---|---|---|---|
| Ioway Creek | 533.58 | 32,765 (32k) | 52 | 5 | 17 |
| Bear Creek | 82.25 | 11,248 (11k) | 10 | 5 | 14 |
| La Maria Creek | 62.41 | 6010 (6k) | 1.5 | 6 | 14 |

Figure 4 presents the obtained meshes and the histogram of the number of faces in the polygons. The number of faces is highly relevant as it is a proxy of the expected model stability during its execution. Meshes with polygons with a relatively large number of faces tend to decrease execution stability, while meshes with fewer faces increase it. Also, a relatively large number of faces tends to increase the model execution times. GMesh delivered most polygons with around six faces or fewer for the three cases.

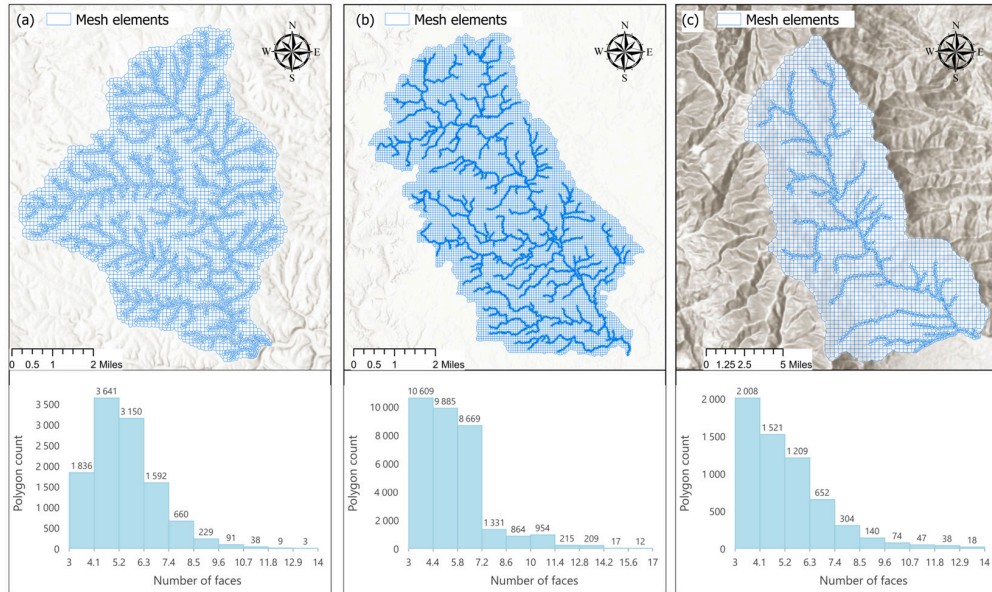

**Figure 4.** Computational grids generated with GMesh. (**a**) for Bear Creek, (**b**) Iowa Creek, and (**c**) La María Creek.

### 3.2. Bear Creek Results

According to the DEM (Figure 3a), the watershed exhibits an elevation gradient of around 200 m with steeper slopes in northern regions close to the watershed boundary. Moreover, the D8 map (Figure 3b) shows a well-defined network. The Bear Creek watershed in northeastern Iowa features rolling hills and steep slopes within the Driftless Area, a rugged region unaffected by glaciation. Its geology includes limestone and dolomite, creating karst features like sinkholes and springs, influencing water flow. Soils derived from loess and alluvium vary from well-drained upland soils to poorly drained lowland soils.

As described in [17], we used the flow direction raster (Figure 3b) to obtain the Voronoi polygons, generating a detailed representation of the watershed's hydrological features. The raster, classified into nine distinct categories, illustrates the various water flow directions within the study area. The DEM (Figure 3a) highlights the topographical variation within the watershed, with elevation values ranging from 245.76 m to 407.22 m.

The model provides a comprehensive overview of the terrain, showcasing high elevation in white and lower elevation in green. This elevation data is essential for understanding the watershed's slope and gradient, which directly impact water flow and erosion processes.

### 3.2.1. Mesh Stability Evaluation

The mesh generation stability process was evaluated in GHOST using three different refinement levels with ten thousand, fifteen thousand, and thirty thousand elements (see Figure 5). The mesh quality was controlled by limiting the polygon's maximum number of faces to 19 and by using a flow accumulation threshold of 300 m$^2$. The results in Table 2 show that the average number of faces is inversely proportional to the number of elements. The average number of faces decreases with many elements, resulting in more stable meshes. Conversely, fewer elements increase the average number of faces, leading to more unstructured meshes and unstable GHOST executions. On the other hand, a larger number of polygons increases the mesh generation computational time (see Table 2).

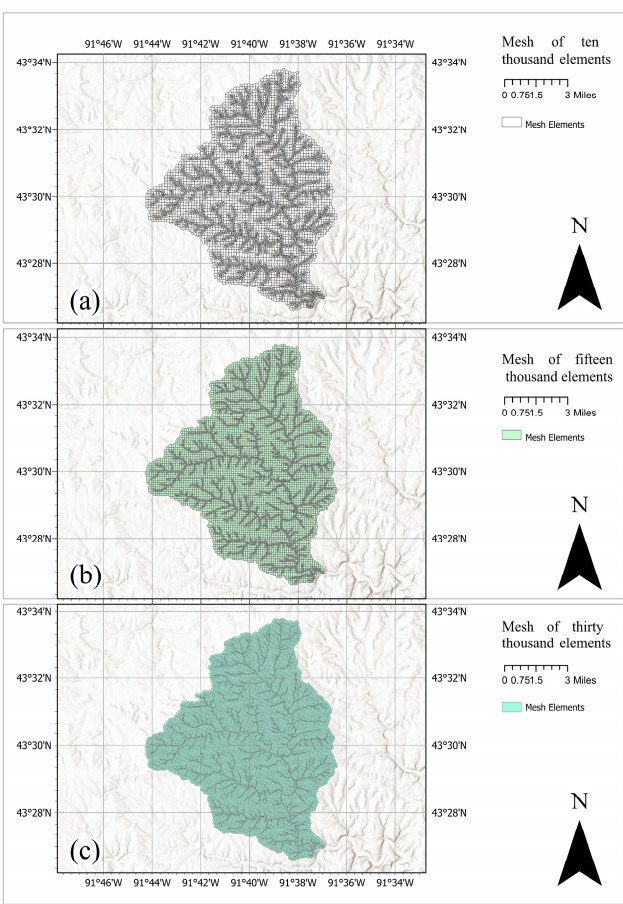

**Figure 5.** Computational grids generated with GMesh. (**a**) Mesh with a thousand elements, (**b**) mesh with fifteen thousand elements, and (**c**) mesh with thirty thousand elements.

**Table 2.** The computational time required to generate the Bear Creek mesh is a function of the number of elements.

| Number of Elements | Generation Time (min) | Average Number of Faces |
| :---: | :---: | :---: |
| 10,000 (10k) | 10 | 11 |
| 15,000 (15k) | 20 | 8 |
| 30,000 (30k) | 35 | 7 |

GHOST was run using the three meshes, obtaining different results at the outlet (Figure 6). According to the results, there is a difference in the peak magnitude and the time to peak between the meshes. The high-resolution mesh (with ~30K elements) exhibits a higher peak flow and a faster response, while the lower resolution (~10K elements) has a lower peak and a slower response. Various authors have reported similar results before. Using the CASC2D-Sed model, ref. [8] showed how coarser DEM resolutions reduced total runoff and peak flow. Moreover, using a finite element hydraulic model [26] shows how the resolution of an irregular triangular mesh reduces the model performance. There is little information on this matter for models like GHOST (physically based and irregular mesh-based). However, the simulated hydrographs indicate that mesh resolution also plays a significant role.

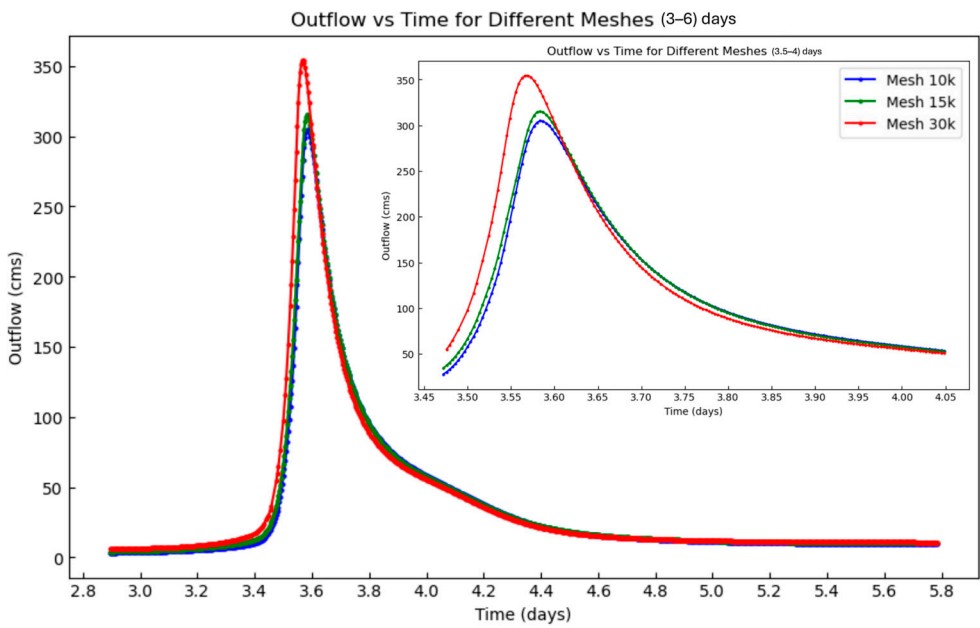

**Figure 6.** GHOST simulation output hydrograph for each mesh resolution and zoom to the simulations between 3.5 and 4 days after the simulation starts.

Higher mesh resolution increased the simulated peak and the response time of the GHOST model. Also, it induced slight changes in the shape of the hydrograph, altering the recession curve and the rising limb. Nevertheless, a higher resolution on the mesh also increased the model execution times (see Table 3). This work explored the mesh resolution effects for a watershed of around 200 km$^2$. However, for larger watersheds, the execution times may increase dramatically, making the high-resolution representation impractical. On the other hand, the resolution effect may change with the watershed scale. We anticipate higher effects over small scales, making the correct selection of the mesh resolution more relevant for these cases. The choice of a proper discretization is of high relevance and can determine the model performance and its parameterization [27–29]. Further studies of this issue using GMesh and GHOST would include the analysis at more scales and under different environments. However, it is out of the scope of this work.

**Table 3.** Bear Creek computational time hydrological model according to the number of elements.

| Number of Elements | GHOST Computational Time (h) |
| --- | --- |
| 10,000 (10k) | 0.5 |
| 15,000 (15k) | 1 |
| 30,000 (30k) | 7 |

According to the results, the computational time to generate the mesh increases proportionally with the number of elements. However, this increase is not linear. For 10,000 elements, the generation time is 10 min, with an average of 11 faces per element. For 15,000 elements, the generation time increases to 20 min, with an average of 8 faces per element. For 30,000 elements, the generation time increases to 35 min, with an average of 7 faces per element.

The increase in generation time with the number of elements is expected due to the higher computational load required to process larger datasets. However, the decrease in the average number of faces per element with increasing elements suggests a more efficient mesh generation at higher resolutions. This efficiency is crucial for detailed hydrological modeling, allowing for more precise simulations without excessively prolonging computation times.

### 3.2.2. Local Mesh Refinement for Bear Creek

To overcome the execution time limitation for large watersheds while representing processes at a relatively high resolution, GMesh has a localized refinement option. The local refinement of the mesh is activated by activating the *focus_map* and *focus_dict* options when calling the *ghost_preprocess* function. The focus_map points to a raster map with categories (numbers) indicating regions with different levels of refinement (e.g., 1, 2, 3, etc.). The *focus_dict* is a Python dictionary where each category has a sub-dictionary with the parameters to build the mesh (segment threshold, mesh distance, segment-to-point distance, etc.). Figure 7 presents an example of this map, indicating three different categories: 1 for the focused areas, 3 for the areas around the channel network, and 2 for the remaining areas.

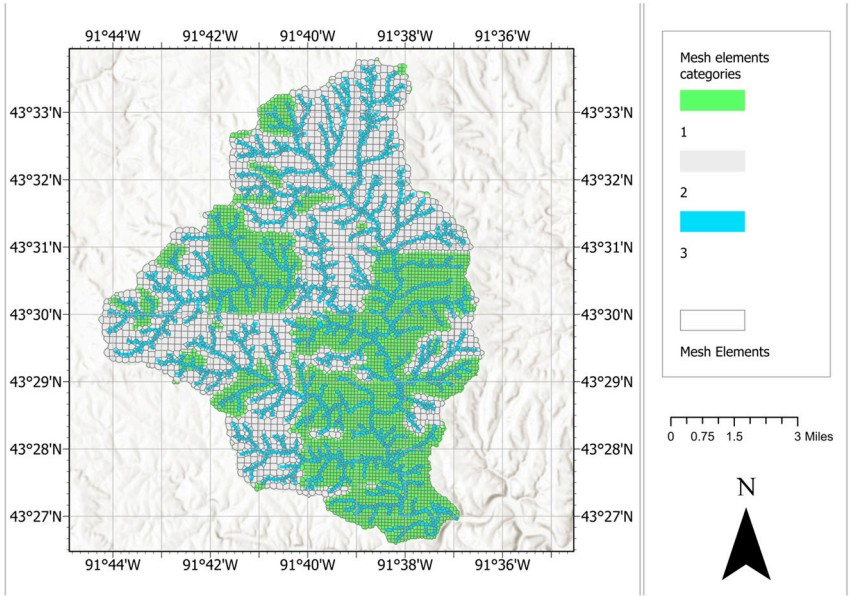

**Figure 7.** Focus mesh generated using the ghost_preprocess WMF module with ten thousand elements using the focus module to create different-sized elements according to zones of interest inside the watershed (Zone 1 = High refinement, Zone 2 = Low refinement, Zone 3 = Medium refinement).

The focus function effectively enhances the mesh resolution in areas of interest, providing a more detailed representation of the watershed. This refinement is crucial for capturing the small-scale hydrological processes that significantly impact the overall simulation accuracy. Also, between the 10K meshes with and without focus areas, the GHOST computational time remains similar (see Table 4).

**Table 4.** Mesh generation and GHOST computational time for a configuration with around 10,000 elements and focus regions.

| Item | Computational Time (h) |
|---|---|
| Mesh generation | 0.2 |
| GHOST execution | 0.4 |

Additionally, the mesh generation cost increases by only two minutes, a reasonable trade-off given the reduction in hydrological computational time.

The hydrograph in Figure 8 illustrates the outflow over time for different mesh configurations, comparing the standard ten thousand (10k) mesh with the ten thousand (10k–focus_mesh) refined areas mesh. The results demonstrate that the locally refined mesh provides a more accurate depiction of the peak flow and recession limb, highlighting the benefits of using the focus function for hydrological simulations. The locally refined mesh captures the hydrological behavior more precisely, particularly in critical areas of the watershed, leading to better predictions of peak flow and recession patterns. Despite the resolution increase, the computational efficiency remains low, demonstrating the practicality of using the focus function for large-scale hydrological modeling.

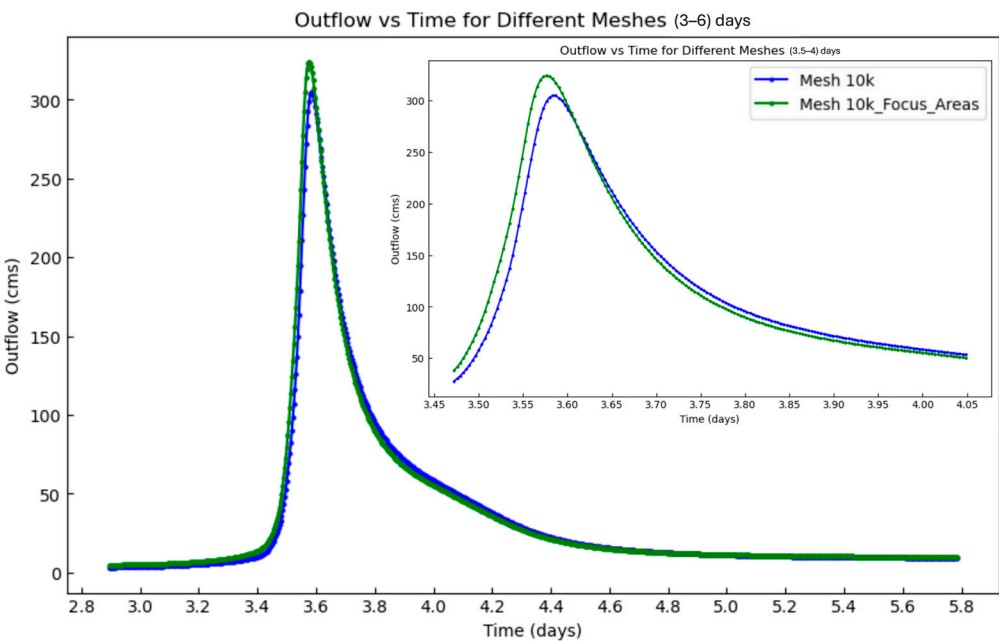

**Figure 8.** GHOST simulation output hydrograph with (green) and without (blue) the focus option activated.

### 3.3. Iowa Creek Results

3.3.1. Local Mesh Refinement for Iowa Creek

A similar analysis was performed for Iowa Creek (see Figure 9). In this case, there was a change in the parameters defining the local refinement in areas close to the network elements. As shown through Figure 9a–c, local refinement increased the number of polygons around the rivers while preserving a relatively low number of faces. Moreover, Table 5 presents the generation time, the average number of faces, and the maximum number of faces. According to it, the increase is relatively low, indicating sustained stability if a model uses the meshes.

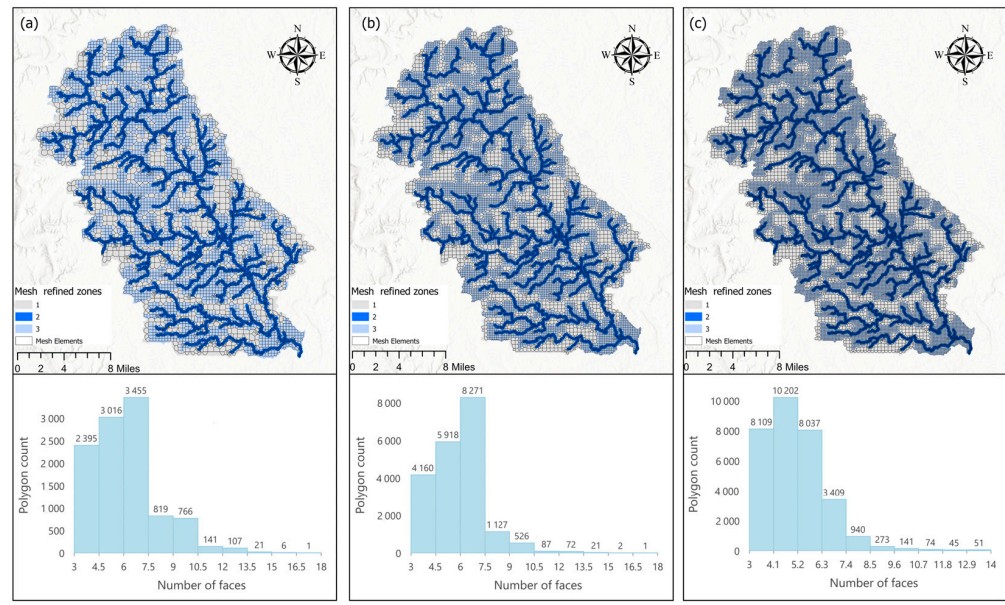

**Figure 9.** Focus mesh generated using the ghost_preprocess WMF module with (**a**) ten thousand elements, (**b**) twenty thousand elements, and (**c**) thirty thousand elements using the focus module to create different-sized elements according to zones of interest inside the watershed (Zone 1 = High refinement, Zone 2 = Low refinement, Zone 3 = Medium refinement).

**Table 5.** Computational time required to generate different zone-refined meshes in Ioway Creek.

| Number of Elements | Generation Time (min) | Average Number of Faces | Max Number of Faces |
|---|---|---|---|
| 10,726 (10k) | 15 | 6 | 18 |
| 20,184 (20k) | 65 | 6 | 18 |
| 31,281 (31k) | 118 | 5 | 14 |

### 3.3.2. Streamflow Simulation Validation

In addition to testing the mesh generation for Iowa Creek, we also tested the GHOST model performance at the outlet of the watershed during a 50-year return period event (see Figure 10). According to the figure, the model achieved a satisfactory performance for twenty thousand elements refinement within a 4 h computational time, a peak flow difference of around 5.7% and a time to peak difference of 0 h.

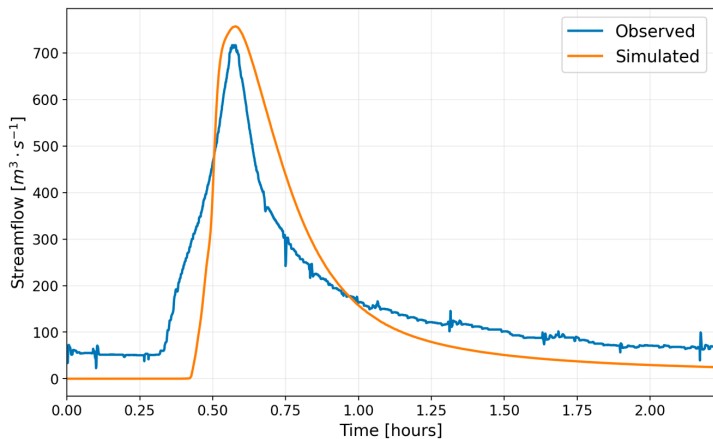

**Figure 10.** Observed and simulated streamflow for a 50-year return period at the outlet of the Iowa Creek watershed. The observations correspond to the USGS gauge 05470500.

*3.4. Comparison with Pyflowline*

Figure 11 compares the mesh generated with GMesh and pyflowline utilizing a watershed near the Susquehanna River at Unadilla, NY, based on an author's example using Jupiter Notebook, created by implementing pyflowline [30]. GMesh provides more flexibility regarding stream segment definition and the option to use local refinement. GMesh provides a balance and flexibility in refinement, integrating topological properties selection within the ghost preprocessor package.

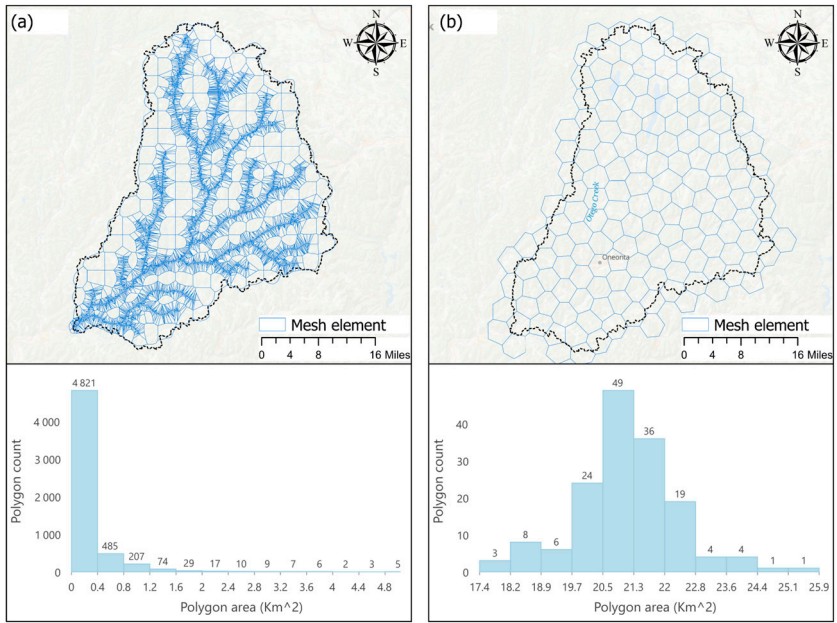

**Figure 11.** Mesh generation using (**a**) GMesh and (**b**) Pyflowline. The bottom histograms present the distribution of the area of the polygons.

## 4. Discussion

The development of GMesh tackles the hydrological modeling challenge of generating computational meshes that accurately capture hydrological features while remaining computationally efficient. Traditional mesh generation approaches often require substantial manual work, are disconnected from watershed hydrological characteristics, and are time-consuming [6]. We designed GMesh to address these issues by automating the process and enabling the direct use of Digital Elevation Models (DEM), flow direction maps, and river network topology. This integration, frequently overlooked in existing tools [11,13], streamlines mesh generation and ensures hydrologically consistent meshes.

We present different GMesh examples using the Bear Creek, Iowa Creek, and La Maria Creek watersheds, highlighting their capacity to generate meshes under different terrain conditions. Simulations using higher-resolution meshes showed differences in the peak flow predictions and faster response times. Our results align with previous studies on the sensitivity of hydrological models to mesh resolution [8,26]. The differences stem from a more detailed representation of surface–subsurface interactions and terrain features. Additionally, the local mesh refinement allowed for assigning computational resources to key areas of the watershed, yielding detailed simulations without significantly increasing overall computation time. This feature offers a practical advantage over traditional uniform refinement methods [22,23]. Moreover, it bridges the gap between computational modeling and real-world hydrological processes. Variable refinement enables targeted simulations, particularly when working with large watersheds or under computational constraints.

This approach aligns with the need for balanced model complexity and efficiency, as emphasized by [15].

Despite these advancements, certain limitations warrant further exploration. While mesh generation time grows with the number of elements, it does not scale linearly, leaving room for optimization. Additionally, the accuracy of generated meshes is highly dependent on input data quality, especially the DEM and flow direction maps. Although GMesh was developed with the GHOST model in mind, extending its compatibility with other hydrological models would broaden its applicability.

Although the change in the resolution and the local refinement presented distinct outflows, there is no information regarding their change in performance. Additional experiments are needed to test the effects of the discretization scale on the model performance. This work presents results at the outlet of the watershed without performing comparisons with observations or at nested sub-watersheds. Previous experiments highlight how network aggregation can blur model assessment downstream [31]. Given the results presented in this work, it is hard to determine the impact of the mesh resolution at smaller channel reaches. Nevertheless, GMesh allows for the performance of the required systematic evaluations. Moreover, it allows robust assessments of the mesh refinement and configuration effects on modeling performance, stability, and computational time.

The comparison of different mesh resolutions underscored the importance of balancing accuracy and computational efficiency. We suspect that higher-resolution meshes provided more accurate predictions of hydrological behaviors, while the ability to refine specific areas within the watershed allowed for targeted simulations of critical processes. Nevertheless, the main goal is to illustrate the versatility of GMesh when implementing and testing different hydrological model configurations. Moreover, its integration with the Google Earth Explorer (GEE) API further enhances its flexibility, making it a comprehensive tool for watershed hydrology.

### 4.1. GMesh Advantages and Limitations

GMesh is a specialized mesh generation software designed for hydrological modeling. It has been built on top of the WMF, allowing interactive access through Python and easy customization of its functionalities. Despite being tailored for the GHOST model, GMesh offers the following advantages:

- In contrast with other available tools, GMesh preserves hydrological features, distinguishing between network and hillslope elements. Additionally, it identifies the connectivity between them.
- GMesh open license and the way it presents the information using known Python variables such as arrays and dataframes, allowing for easy customization.
- GMesh allows for the definition of different levels of refinement within the same project.
- The GMesh connectivity with Google Earth Engine (GEE) allows an easy retrieval of land use and soil properties, enhancing the implementation of the model in different regions.
- Once executed, GMesh writes the files required for GHOST and writes the vector maps of the watershed, including the mesh and the network. Moreover, GMesh allows the definition of the variables that will be contained in the vector layers.

Nevertheless, GMesh also has some limitations that may impact its implementation under different circumstances. Some of these limitations are:

- While optimized for the GHOST model, adapting GMesh for use with other hydrological models may require additional customization.

- High-resolution meshes can lead to increased computational requirements, potentially limiting scalability for extensive watershed analyses. However, this issue applies to other mesh generators and to hydrological PDE models where modelers need to define relatively large simulation elements.
- GMesh execution time could be improved by implementing parallel approaches, adapting the usage of graphics processing units (GPUs), and migrating some of its code to Fortran or C (as has been done in the WMF).

### 4.2. Comparison with Similar Tools

Currently, there are similar tools that also offer mesh generation for different purposes. GMesh stands out from general-purpose tools like Gmsh [13] by offering hydrologically informed mesh generation, preserving river–hillslope connectivity and watershed structure. While Gmsh and FEATool [32] are versatile, they lack built-in support for hydrological features. On the other hand, ADMESH+ [16] and PyFlowline [33] offer partial solutions—focusing on hydraulic grids or river networks—but do not provide full watershed meshing. Finally, unlike commercial tools such as HydroGeoSphere [11] or HYDRUS 3D [12], GMesh is open-source, scriptable, and optimized for automation. In Table 6, we present some of the most popular mesh generation tools, their description, advantages, limitations, and licenses.

**Table 6.** Comparison of different mesh generation tools with potential application to hydrological modeling.

| Tool | Description | Advantages | Limitations | License |
|------|-------------|------------|-------------|---------|
| GMesh | Automated watershed-oriented mesh generator | Preserves river-hillslope connectivity, local refinement, and support GEE support | Currently tied to GHOST, computationally intensive | GNU V3.0 |
| Gmsh | General-purpose 3D finite element mesh generator | Versatile, GUI interaction, community support | Not tailored for hydrology and requires manual integration | GPL |
| ADMESH+ | Mesh generation for 1 and 2D hydrodynamic models. | Integrates DEM, land-water data and refines around hydrological features | Focused on hydraulics and is less flexible for hydrological simulations | MIT |
| PyFlowline | Mesh generation for hydraulic simulations | River-network-focused and works with structured/unstructured meshes | Is it not a full mesh generator. Supplies riverine data | MIT |
| FEATool Multiphysics | Integrated modeling tool with support for multiple physics including flow and mesh generation | Has a GUI and is scriptable, supports several physics and exports to OpeanFOAM and COMSOL | General purpose not focused on hydrological modeling | Free with paid upgrades |
| HydroGeoSphere | Fully integrated 3D physically based hydrological model | It is well tested for soil moisture and ground water modeling | Has commercial limitations | Free for 1/2D and paid for 3D |

**Table 6.** *Cont.*

| Tool | Description | Advantages | Limitations | License |
|------|-------------|------------|-------------|---------|
| D-Flow Flexible Mesh [14] | Unstructured mesh generator for hydrodynamics | High-quality GUI, integration with Deltf3D products, and has coastal and riverine applications | It is GUI-focused, paid for advanced features, and not open | Freemium/Paid |
| DIVA [34] | Spatial gridding and interpolation tool for coastlines and sub-basins | Good for marine/coastal applications | Not a mesh generator per se | GNU V3.0 |
| Hydrus 3D | Software package for simulating water, heat, and solute movement in 3D variably saturated media | It is well tested for soil moisture and ground water modeling | Is it not specialized in hydrological network structure | Free for 1/2D and paid for 3D |

As shown in the table below, GMesh fills a gap in the current available tools for mesh generation tailored towards the implementation of hydrological models. Some tools, like HydroGeoSphere or Hydrus3D, are also developed for hydrology. However, these are usually closed and require the usage of their specific hydrological model. Currently, GMesh also faces a similar limitation. Nevertheless, its license and scripted approach will allow the community to expand its application to other models.

Future work should focus on evaluating GMesh across diverse watershed scales and environments to assess its generalizability. Incorporating real-time data streams and developing adaptive refinement strategies responsive to dynamic hydrological conditions could further enhance forecasting applications. There is also potential to improve computational performance through parallelization or leveraging GPU processing, particularly for large-scale simulations.

## 5. Conclusions

This study introduced GMesh, an automated watershed-oriented mesh generator integrated with the Watershed Modeling Framework (WMF) for use in physically based hydrological models. GMesh streamlines the traditionally labor-intensive process of mesh generation, allowing for the efficient creation of computational meshes with varying levels of detail. This work presents examples of the GHOST model running GMesh-generated meshes, demonstrating its ability to obtain stable and structured meshes. The flexibility offered by GMesh, particularly its ability to perform local refinement, makes it a powerful tool for hydrological simulations requiring detailed and iterative processes. The Supplementary Materials section presents the links to the software and the data required to test it.

GMesh achieves this by integrating digital elevation data (DEM) and flow direction maps to delineate the watershed structure and identify the channel network. The software then generates mesh points around these network elements using Voronoi polygons, ensuring that the spatial relationships between land and river segments are preserved. This process creates a mesh where each polygon is oriented toward hydrological processes, with the ability to capture both the terrain's topographical characteristics and the flow dynamics of the watershed. By automating the recognition and connectivity of land and network elements, GMesh produces meshes inherently suited for physically based hydrological simulations, ensuring accuracy in surface–subsurface interaction modeling.

GMesh was validated by applying it to the Bear Creek watershed, Iowa (200 km$^2$). The validation consisted of experiments showing the GMesh capabilities and how it can benefit the implementation of finite element-based hydrological models. The work starts by presenting a straightforward implementation using the same level of refinement. Then, it evaluated GMesh stability by generating meshes of different refinement levels for Bear Creek. Finally, it presented an example of how the local refinement works. For the refinement levels, three meshes were shown with a total number of elements varying between 10 and 30 K. In this case, changes in the mesh generation time and the average number of faces of the generated polygons were presented. Also, a contrast between GHOST simulations was discussed in which the highest refinement level provided the most significant peak flows. In addition, the work described the differences between simulations with and without local mesh refinement (Section 3.3).

However, despite these strengths, GMesh has several limitations that warrant further development. The computational time required for generating high-resolution meshes increases significantly. Additionally, the accuracy of the mesh is highly dependent on the quality of input data, such as the DEM, and uncertainties in the flow direction maps. While GMesh allows for local refinement, further optimization is needed to improve its performance in regions with complex topography. Finally, the current version of GMesh is primarily compatible with the GHOST model. Future work may explore expanding compatibility and optimizing its compatibility with other models.

In summary, GMesh provides a practical solution for hydrologists seeking to generate high-quality meshes efficiently. Automating a traditionally labor-intensive process and preserving key hydrological connections enables more accurate, flexible, and computationally manageable simulations. This tool serves researchers and practitioners aiming to improve hydrological analysis and decision-making processes.

**Supplementary Materials:** The pre-processing code is included as a branch of the Watershed Modelling Framework (WMF) in GitHub and can be found at: https://github.com/nicolas998/WMF/tree/ghos_topo (accessed on 15 September 2025). This free and open software has been available since 2021, written in Fortran and Python. It requires a Python 3.X interpreter and a Fortran compiler (we tested it using gFortran). For a complete description of each function and examples of their usage, please refer to: https://github.com/nicolas998/WMF/blob/develop/Examples/BearCreek_Watershed_Oriented_Mesh_Generator.ipynb.

**Author Contributions:** Conceptualization, N.V. and A.A.; methodology, N.V.; software, N.V. and M.D.; validation, M.D.; formal analysis, M.D. and N.V.; investigation, N.V., A.A. and M.D.; resources, N.V. and A.A.; data curation, M.D.; writing—original draft preparation, N.V.; writing—review and editing, M.D. and A.A.; visualization, M.D. and N.V.; supervision, N.V. and A.A.; project administration, N.V. and A.A.; funding acquisition, N.V. and A.A. All authors have read and agreed to the published version of the manuscript.

**Funding:** This research was partially funded by the Mid-America Transportation Center via a grant from the U.S. Department of Transportation's University Transportation Centers Program (USDOT UTC grant for MATC: 69A3551747107), the Iowa Highway Research Board, and the Iowa Department of Transportation (project: TR-699), and the Iowa Water Approach Project, https://iowawatershedapproach.org/.

**Data Availability Statement:** The maps and time series of the examples presented in this work can be found in the following Zenodo repository: Link: https://zenodo.org/records/14947022?token=eyJhbGciOiJIUzUxMiIsImlhdCI6MTc0MDc2MzY0MCwiZXhwIjoxNzY3MTM5MTk5fQ.eyJpZCI6IjJiZTY2YjU4LThjMGItNDI5NC05ZTkwLWUxZjI2ZGFlN2JhMiIsImRhdGEiOnt9LCJyYW5kb20iOiI4MTg2YjU1NzRlYmVkMjQzYmMxMjc2YTU1NjQ1Y2JiNSJ9.WeChUlqbUmN0yd-SX6SGGrY_NQrbOVEQz8YGSusYwH2oOFJRbpmbztfS7zliQgGuObXprsBBSSuwRBrh2qZNWQ (accessed on 17 September 2025).

**Acknowledgments:** This work was completed with partial support from the Iowa Flood Center.

**Conflicts of Interest:** The authors declare no conflicts of interest.

## Abbreviations

The following abbreviations are used in this manuscript:

| | |
|---|---|
| GHOST | Generic Hydrologic Overland-Subsurface Toolkit |
| WMF | Watershed Modelling Framework |
| GMesh | GHOST Mesh generator |

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
