# Peer review of "GMesh: A Flexible Voronoi-Based Mesh Generator with Local Refinement for Watershed Hydrological Modeling"

_hydrology, doi:10.3390/hydrology12100255_

Round 1
Reviewer 1 Report (Previous Reviewer 1)
Comments and Suggestions for Authors
Further, i don't have any comment.
Reviewer 2 Report (Previous Reviewer 2)
Comments and Suggestions for Authors
The authors addressed the comment. Now the paper is ready for the decision.
This manuscript is a resubmission of an earlier submission. The following is a list of the peer review reports and author responses from that submission.
Round 1
Reviewer 1 Report
Comments and Suggestions for Authors
General Comments
- Please remove the word “We” from the whole manuscript.
- Include one figure a table that shows your input of model development in terms of time reduction or ……
- The study discusses the effect of mesh resolution on hydrological modeling outputs but does not compare the simulation results with real observed streamflow or hydrological data.
- The study does not explore how sensitive the results are to different model parameters, such as DEM resolution, flow accumulation threshold, or mesh refinement strategy.
- The paper discusses computational time for mesh generation and hydrological simulations but does not compare performance against existing mesh generators.
- The study is limited to Bear Creek Watershed (~200 km²), but does not explore whether GMesh performs well in larger or more complex watersheds.
- The study does not quantify uncertainties in mesh generation or hydrological simulations.
- The results show that higher-resolution meshes lead to different peak flow predictions, but it is unclear why????????
- The study briefly mentions future improvements, but does not outline concrete steps.why???
Specific Comments
Introduction
Line 29-30
The citation style is incorrect [1] and [2].
Line 43
Typos error .”
There is a repetition in the word “bottleneck” please avoid from the repetition
Line 53-58
Again repetition of “We develop….., We built….” This is inappropriate. Please remove it.
Line 61-62
Grammatical error “Gmesh allows generation mesh” please check it again
Materials and Methods
Line 74
Very poor way of writing “as shown in [17]”
Line 76
In the whole manuscript only one time you used “TETIS” no one knows about this model. Could you please explain it in a way that the reader can easily understand .
Line 242-243
Please make a table in which you should make a comparison of mesh cell size with the previously published data. Also the include time taking to do a mesh.

English is good, however, it should be improved
Reviewer 2 Report
Comments and Suggestions for Authors
This research introduces GMesh, an automated mesh generator designed for hydrological modeling within the Watershed Modeling Framework (WMF). GMesh enhances the efficiency of mesh generation by allowing for customizable configurations and local refinements, which improve the accuracy of hydrological simulations. The study validates GMesh through its application to the Bear Creek watershed, demonstrating its capability to produce high-resolution meshes that significantly impact peak flow predictions and model performance. However, there are major flaws that need to be addressed before the final decision:
- The current version of GMesh shows limitations in performance when applied to regions with complex topography. Addressing this issue through further optimization could enhance the reliability of the mesh generation process, ensuring that it can effectively handle diverse geographical features.
- he accuracy of the generated meshes is highly dependent on the quality of input data, such as DEMs and flow direction maps. To improve reliability, the research should explore methods to assess and enhance the quality of these input datasets, as uncertainties in them can significantly affect model outcomes.
- GMesh is primarily compatible with the GHOST model, which limits its applicability for users of other hydrological models. Expanding its compatibility to include a wider range of models would not only broaden its usability but also enhance the overall reliability of the research by allowing for more comprehensive testing and validation across different hydrological frameworks
Specific
- Title is neither clear (or support objectives) nor contains methodlogy part – revise it
- Introduction: Research objectives are not mentioned clearly. Research Gap is not presented scientifically, there are major recent literature have been missed add those from 2025 and 2024 (now it is limited) - What is not known? What elements are still subject to controversy? What is the exact gap in the knowledge that your study hopes to fill? Cite any existing data, especially conflicting data that indicate uncertainty. Structure of the paper should be written clearly
- Method section is poorly drafted; The objective of the methods section is to describe exactly what you did, and how, in sufficient detail such that any average reader with the same resources at their disposal would be able to reproduce your study. There must be a method described for every result you intend to include in your results section – i.e., you cannot present the results of a test or analysis that was not mentioned in the methods.
Give full information of the dataset fetched with the date of data accessed. Resolution, link etc. in tabulated form.
Give full detail of the applied approach with true citation – so reader can fetch further detail,
Mathamatical expression of those applied methods are missing.
- Results is poorly drafted now, and limited – present what has been observed, and report all without selective reporting.
- Conclusion: this is elaborated form, kindly compact it, give the most relvent information as a conclusion including future research objectives.
- Abbreviation is poorly reported, define abbreviation at very first use.
- English needs serious revision – the scientific writing is missing, context is not scientifically presented at many places.
Comments on the Quality of English Language
- English needs serious revision – the scientific writing is missing, context is not scientifically presented at many places.